# Unique Polyhalogenated Peptides from the Marine Sponge *Ircinia* sp.

**DOI:** 10.3390/md18080396

**Published:** 2020-07-28

**Authors:** Rogelio Fernández, Asep Bayu, Tri Aryono Hadi, Santiago Bueno, Marta Pérez, Carmen Cuevas, Masteria Yunovilsa Putra

**Affiliations:** 1Natural Products Department, PharmaMar S.A., Pol. Ind. La Mina Norte, Avda. de los Reyes 1, 28770 Colmenar Viejo (Madrid), Spain; rfernandez@pharmamar.com (R.F.); sbueno@pharmamar.com (S.B.); ccuevas@pharmamar.com (C.C.); 2Research Center for Biotechnology, Indonesian Institute of Sciences, Jl. Raya Jakarta-Bogor No. Km46, Cibinong, Bogor, Jawa Barat 16911, Indonesia; asepbayu86@gmail.com (A.B.); masteria.yunovilsa@gmail.com (M.Y.P.); 3Research Center for Oceanography, Indonesian Institute of Sciences, Jl. Pasir Putih I, Ancol Timur, Jakarta 14430, Indonesia; tri.aryono.hadi@lipi.go.id

**Keywords:** marine sponge, *Ircinia* sp., polyhalogenated peptides, Marfey’s analysis

## Abstract

Two new bromopyrrole peptides, haloirciniamide A (**1**) and seribunamide A (**2**), have been isolated from an Indonesian marine sponge of the genus *Ircinia* collected in the Thousand Islands (Indonesia). The planar structure of both compounds was assigned on the basis of extensive 1D and 2D NMR spectroscopy and mass spectrometry. The absolute configuration of the amino acid residues in **1** and **2** was determined by the application of Marfey’s method. Compound **1** is the first dibromopyrrole cyclopeptide having a chlorohistidine ring, while compound **2** is a rare peptide possessing a tribromopyrrole ring. Both compounds failed to show significant cytotoxicity against four human tumor cell lines, and neither compound was able to inhibit the enzyme topoisomerase I or impair the interaction between programmed cell death protein PD1 and its ligand, PDL1.

## 1. Introduction

Indonesia is located at the center of a biodiversity hotspot, and around 750 structures from Indonesian waters have been published in the last 50 years [1]. The structural diversity and bioactive properties of the compounds isolated from this region encouraged us to continue to investigate this area, which still remains largely unexplored. Thus, one of the recent PharmaMar expeditions was carried out in the Thousand Islands, in collaboration with Indonesian Institute of Sciences (LIPI). The Thousand Islands archipelago is located about 25 miles from the coast to the northeast of Jakarta; the collection site is an area full of gentle rock and coral slopes, and it is potentially a highly productive area, both in terms of the quantity and the nature of the biodiversity. In this paper, we describe the isolation of two new peptides isolated from an *Ircinia* specimen from this area.

Of all the marine organisms investigated, sponges (Porifera) are the most primitive multicellular animals with ample time to evolve into more complex living organisms. In fact, marine sponges are recognized as the richest sources of MNP, contributing to nearly 30% of all marine natural products discovered so far [2]. Previous reports revealed that Marine sponges of the genus *Ircinia* are known as a rich source of varied bioactive natural products, including fatty acids [3], steroids [4,5], terpenes [6,7], macrolides [8,9], and peptides [10], many of which have biological activities. This structural diversity could be due to the fact that sponges harbor diverse microorganisms and in numerous cases, bacteria isolated from sponges or symbiotic bacteria are the true producers of the compounds found in their extracts [11]. Specifically, an intriguing group of *Ircinia*-derived peptides are assumed to be of microbial origin due to the presence of both d-amino acids and unusual amino acids, as illustrated by the cyclic hexapeptide waiakeamide from *Ircinia dendroides* [12] and the cyclotheonamides E4 and E5, which are cyclic pentapeptides also from *Ircinia* species [13].

Undeniably, among the compounds isolated from marine sources, linear and cyclic peptides are recognized as an important class with great structural diversity and a wide range of bioactivities, and these include the antimalarial carmabin A [14], the antiproliferative jaspamides [15], and the cytotoxic patellamides [16]. Furthermore, two marine peptide-derived products have reached the market: ziconotide [17] for analgesic use and a synthetic derivative of dolastatin 10 [18] linked to an antibody for the treatment of Hodgkin’s lymphoma. PharmaMar has also developed a marine natural peptide Aplidin, which was originally found in the Ascidian *Aplidium albicans* and has recently been approved for commercialization in Australia for the treatment of multiple myeloma. Recent studies suggest that Aplidin may also have antiviral properties, and a clinical trial to treat patients with COVID-19 has been initiated.

In the course of our screening program to isolate novel compounds with antitumor properties from marine sources, the organic extract of an *Ircinia* specimen collected off the coast of the Thousand Islands showed hints of activity, and although the fractions did not confirm cytotoxicity, the chromatographic profiles along with the mass spectra showed peaks that were interesting enough for us to purify them. Thus, we have isolated two unique peptides haloirciniamide A (**1**) and seribunamide A (**2**). It is worth mentioning that the number of known peptides with a halogenated pyrrole ring is limited, with only cyclocinamides and corticiamide A [19] as well as gunungamide A [20] having been described as possessing chlorinated pyrrole rings. Although there are dozens of dibromopyrrolecarboxamide derivatives from porifera such as nagelamide [21] and carteramine [22], mainly from *Agela* and *Stylissa* species, compound **1** is the first example of a peptide containing a halogenated pyrrole ring in its structure. Indeed, haloirciniamide A represents a structurally unique cyclopeptide, since it also has an unprecedented chlorohistidine moiety. Furthermore, there are only two examples of tribromopyrrole rings derived from natural sources: 2,3,4-tribromopyrrole itself, which was isolated from the marine Poychaete *Polyphysia crassa* [23], and tribromopyrrol-2-methylphenol isolated from a coralline algal-associated *Pseudoalteromone* [24], with compound **2** being the first of its class.

Details of the isolation and structural elucidation of the new halogenated peptides **1** and **2** are provided. The results of cytotoxicity and other related antitumor programmed cell death protein (PD1) and TOPO-I screenings are also described.

## 2. Results and Discussion

### Isolation and Structure Elucidation

The sponge *Ircinia* sp. was collected by hand while diving in the Thousand Islands (Indonesia). The specimen was repeatedly extracted using CH_2_Cl_2_:MeOH (1:1 *v*/*v*). The combined concentrated extracts, after vacuum liquid chromatography (VLC) and semipreparative reverse-phase HPLC separations, led to the isolation of the two pure compounds shown in Figure 1.

Compound **1** was isolated as an amorphous white solid. The isotopic distribution observed in the (+)-LRESIMS mass spectrum with four protonated ion [M+Na]^+^ peaks at *m/z* 830, 832, 834, and 836 in the ratio 3:6:3:1 respectively, showed the presence of two bromine atoms and a chlorine atom in the molecule. The presence of these halogens in the structure was confirmed by (+)-HRESI-TOFMS analysis, with the ion peak observed at *m/z* 830.0400 [M+Na]^+^ corresponding to the molecular formula C_25_H_32_^79^Br_2_^35^ClN_11_O_8_Na (calcd. 830.0383). Interpretation of the mono NMR data (^1^H, ^13^C and 1D-TOCSY) compiled in Table 1 and two-dimensional NMR spectra (gHSQC, gCOSY, gHMBC, and 2D-TOCSY) in CD_3_OD led to identification of 6 spin systems. Taking into consideration the seven carbonyl carbon resonances (δ_C_ 161.4–175.8) and the number of α-amino acid proton signals (δ_H_ 4.06–4.84), the peptide nature of compound **1** was expected. This hypothesis was confirmed by the NH signals observed in the ^1^H NMR spectrum in CD_3_OH (δ_H_ 7.13–12.68) and DMSO-d_6_ (δ_H_ 7.37–12.03), with the latter solvent being chosen for full structural elucidation. The COSY correlations observed between methines at δ_H_ 4.23/δ_C_ 50.5 and δ_H_ 4.49/δ_C_ 51.5 with the diastereotopic methylenes at δ_H_ 2.87, 3.13/δ_C_ 49.2 and δ_H_ 3.22, 4.04/δ_C_ 40.5 respectively, indicated the presence of two 2,3-diaminopropionic acid units (Figure 2). Both amino acids were directly connected based on the HMBC correlation of the NH signals at δ_H_ 9.11 for Dap1 and δ_H_ 7.72 for Dap2 with the same carbonyl carbon at δ_C_ 169.5. The next amino acid present in the peptide core was an isoserine with a methylene group at δ_H_ 2.75, 3.47/δ_C_ 42.8, a methine carbon at δ_H_ 4.13/δ_C_ 67.8, and an NH signal at δH 8.23. Isoserine was linked to the Dap2 by the correlation between NH at δ_H_ 8.23 and the carbonyl carbon of Dap2 at δ_C_ 170.6. In addition, an HMBC correlation of the methine carbon at δ_H_ 4.13 and the NH signal at δ_H_ 7.13, which belongs to a unit of isoasparagine (δ_H_ 4.58/δ_C_ 48.4, δ_H_ 2.80, 3.09/δ_C_ 35.0), with the carbonyl group at δ_C_ 170.6 allowed the sequence of amino acids to be continued. Finally, the peptide ring was closed by an NMe-histidine (δ_H_ 3.99/δ_C_ 65.6, δ_H_ 3.05/δ_C_ 25.0), whose N-methyl group showed an HMBC correlation with carbons belonging to the carbonyl group of iAsn at δ_C_ 173.5 and its own methine at δ_C_ 65.6, with an additional HMBC correlation between the α-aminoacid proton signal of NMe-histidine and the NH signal at δ_H_ 9.11 of Dap1 with the carbonyl carbon at δ_C_ 169.0 (Figure 2). NMe-histidine ring shifts at δ_C_ 110.5, δ_C_ 128.2, and δ_H_ 6.76/δ_C_ 109.5 revealed that the non-protonated carbon at δ_C_ 128.2 bore one of the three halogen atoms.

To complete the structure elucidation, the two remaining doublets with a small coupling constant value of 2.7 Hz were assigned to an sp^2^ methine at δ_H_ 6.30/δ_C_ 110.4 and a significant downfield NH signal at δ_H_ 12.68. An HMBC correlation of these two protons with three non-protonated sp^2^ carbons (δ_C_ 96.9, 118.1, and 123.2), demonstrated the existence of a trisubstituted pyrrol moiety, with two of these three positions bearing halogens. The placement of this heterocycle was established by the HMBC correlation of the methylene group of Dap2 with a carbonyl group at δ_C_ 158.6. To confirm the direct connection between the pyrrol moiety and this carbonyl group, a new gHMBC experiment with J = 3 Hz was conducted. The position of the sp^2^ methine in the pyrrol unit and the bond to the ring with the cyclopeptide was settled by the HMBC correlation between the methine proton at δ_H_ 6.30 and the carbonyl group at δ_C_ 158.6. Although the chlorine and two bromine atoms were undoubtedly located on the three free positions of the heterocycle ring and the carbon shifts (δ_C_ 96.9 and 123.2) suggested that both bromine atoms were on the pyrrol moiety, this evidence was insufficient to fully confirm this proposal. Fortunately, this could be resolved by detailed study of the peptide structure by (+)-HRESI-TOFMS and QTOFMS (Figure 3), which showed significant cluster ions at *m/z* 158.0487/160.0460 in a 3:1 ratio corresponding to the iAsn moiety. These *m/z* values, the mass error observed, and the isotopic distribution clearly confirmed the presence of a chlorine atom on the NMeHis amino acid (Appendix A).

The absolute stereochemistry of compound **1** was established on the basis of Marfey’s analysis with the 1-fluoro-2,4-di-nitrophenyl-5-l-alanine amide (l-FDAA) [25]. Compound **1** was hydrolyzed in strong acid conditions and derivatization of the free amino acids with l-FDAA allowed an exhaustive analysis by HPLC-MS. A comparison of the retention times of the derivatized amino acids present in **1** and the suitably derivatized pure amino acid standards unambiguously demonstrated the absolute configuration as l-Dap, l-iSer, and d-Asn. The absolute configuration of NMeClHis could not be determined due to the absence of the standard amino acid.

Compound **2** was isolated as an amorphous white solid. Its (+)-LRESI showed an *m*/*z* = 825 [M + H]^+^ with a characteristic cluster corresponding to the presence of three bromine atoms. The molecular formula C_29_H_43_^79^Br_3_N_6_O_7_ was established by (+)-HRESI-TOFMS analysis of the [M + H]^+^ at m/z 825.0806 (calcd. 825. 0816). The peptide nature of **2** was evident from its ^1^H and ^13^C NMR spectra (Table 2). ^1^H NMR in DMSO showed the characteristic α-proton resonances of four α-amino acids in the range δ_H_ 5.35 to 4.10 ppm, five interchangeable protons at δ_H_ 12.53, 8.56, 8.08, 7.19, and 6.77 ppm, and two NMe signals at δ_H_ 3.61 and 3.04 ppm. ^13^C NMR data displayed six carbonyl signals between δ_C_ 173.5 and 159.1 ppm, four adjacent methine carbons in the range δ_C_ 59.1–51.5 ppm, and two NMe signals at δ_C_ 35.7 and 30.5 ppm. Extensive 2D NMR analysis, including COSY, TOCSY, HSQC, and HMBC was used to determine the identity of the four amino acids and to assign the NMR signals. As a result of these studies, the amino acids were found to be one Ile, one NMe-Leu, one Pro, and one Asn unit. A long-range correlation between protons at δ_H_ 7.19/6.77 and 2.15/2.11 ppm with the carbonyl group at δ_C_ 173.5 ppm and the observation of ROESY cross-peaks between protons at δ_H_ 7.19/6.77 ppm and the CH_2_ of position 4 at δ_H_ 2.15/2.11 ppm established the presence of a Gln. A N-methyl-2,3,4-bromopyrrol unit was inferred by the presence of four aromatic non-protonated carbons at δ_C_ 128.5, 107.9, 100.7 and 93.4 ppm with chemical shifts similar to those described for bromopseudoceratines [23].

The sequencing for compound **2** was carried out using a combination of HMBC and ROESY data. Long-range correlations from α-protons, NH, and NMe to carbonyl carbons of adjacent amino acids plus ROESY correlations between α-protons, NH, and NMe protons of adjacent amino acids (see Figure 2) allowed us to establish the sequence as Br_3_Py-Ile-NMeLeu-Pro-Gln.

The absolute configurations of the amino acids were determined by comparing the hydrolysis products of **2** (6 N HCl, 110 °C, 18 h) after derivatization with Marfey’s reagent (N-(3-fluoro-4,6-dinitrophenyl)-l-alaninamide, l-FDAA), with appropriate amino acid standards using HPLC-MS chromatography. As a result, all the amino acids were determined to be *L*.

## 3. Materials and Methods

### 3.1. General Experimental Procedures

Optical rotations were determined using a Jasco P-1020 polarimeter. UV spectra were performed using an Agilent 8453 UV−vis spectrometer. IR spectra were obtained with a Perkin-Elmer Spectrum 100 FT-IR spectrometer with ATR sampling. NMR spectra were recorded on a Varian “Unity 500” spectrometer at 500/125 MHz (^1^H/^13^C). Chemical shifts were reported in ppm using residual CD_3_OH (δ 3.31 ppm for ^1^H and 49.0 ppm for ^13^C) and DMSO-d_6_ (δ 2.50 ppm for ^1^H and 39.5 ppm for ^13^C) as an internal reference. HRESI-TOFMS was performed on an Agilent 6230 TOF LC/MS chromatograph spectrometer. (+)-ESIMS were recorded using an Agilent 1100 Series LC/MSD spectrometer. HRESI-TOFMS was performed on an Agilent 6230 TOF LC/MS chromatograph spectrometer. ESI(+) and MS^e^ were performed on an Waters UHPLC-QTOF Acquity I-Class + Xevo G2-XS.

### 3.2. Biological Material

The sponge *Ircinia* sp. (158 g) was collected by hand using a diving rebreather system in the Thousand Islands (Indonesia). The sponge was immediately frozen and kept under these conditions until extraction. The specimen was identified by María Jesús Uriz at CEAB, Blanes, Spain. A voucher specimen (ORMA155272) was deposited at PharmaMar facilities (Madrid, Spain).

### 3.3. Extraction and Isolation

The sponge *Ircinia* sp. (158 g) was triturated and exhaustively extracted with MeOH:DCM (1:1, 3 × 500 mL). The combined extracts were concentrated to yield a crude mass of 7.9 g. The crude product was subjected to VLC on a Lichroprep RP-18 with a stepped gradient from H_2_O to MeOH to CH_2_Cl_2_. The fractions eluting with H_2_O:MeOH (3:1, 606 mg) and H_2_O:MeOH (1:1, 89.6 mg) were subjected to semipreparative HPLC (Symmetry Prep C_18_ 5 μm, 10 × 150 mm; 3 min isocratic H_2_O + 0.04% TFA: CH_3_CN + 0.04% TFA 95:5 and then gradient from 5% to 68% CH_3_CN + 0.04% TFA in 25 min, flow 3 mL/min, UV detection) to obtain 4.3 mg of compound **1**. The fraction eluting with H_2_O:MeOH (1:3, 43.9 mg) was subjected to semipreparative HPLC (Symmetry Prep C_18_ 5 μm, 10 × 150 mm; 3 min. isocratic H_2_O + 0.04% TFA: CH_3_CN + 0.04% TFA 90:10 and then gradient from 10% to 75% CH_3_CN + 0.04% TFA in 25 min, flow 3 mL/min, UV detection) to obtain 3.0 mg of compound **2**.

Haloirciniamide A (**1**): amorphous white solid; [α]^25^_D_ –62.7º (c 0.1, MeOH); IR υmax 3314, 2920, 2850, 1644, 1523, 1416, 1311, 1239, 1199, 1041 cm^−1^; UV (MeOH) λ_max_ 198, 268 nm. ^1^H NMR (500 MHz) and ^13^C NMR (125 MHz) see Table 1; (+)-HREI-TOFMS *m/z* 830.0400 [M+Na]^+^ (calcd for C_25_H_32_^79^Br_2_N_11_O_8_Na *m/z* 830.0383).

Seribunamide A (**2**): amorphous white solid; [α]^25^_D_ –38.4º (c 0.2, MeOH); IR υmax 3352, 2932, 2850, 1658, 1515, 1320, 1236, and 1035 cm^−1^; UV (MeOH) λ_max_ 197, 266 nm. ^1^H NMR (500 MHz) and ^13^C NMR (125 MHz) see Table 2; (+)-HREI-TOFMS *m/z* 825.0806 [M + H]^+^ (calcd for C_29_H_44_^79^Br_3_N_6_O_7_
*m/z* 825.0816).

### 3.4. Absolute Configuration

Absolute Configuration of **1**. First, 0.5 mg of haloirciniamide A was hydrolyzed in 0.5 mL of 6 N HCl at 110 °C for 15 h. The excess aqueous HCl was removed under a N_2_ stream, and a solution of 700 µg of l-FDAA (N-(3-fluoro-4,6-dinitrophenyl)-l-alanine-amide) in acetone (160 µL), H_2_O (100 µL), and NaHCO_3_ 1N (50 µL) was added to the dry hydrolysate. The resulting mixture was heated at 40 ºC for 1 h, before being cooled to 23 ºC, quenched by addition of 2N HCl (20 µL), dried, and dissolved in H_2_O (800 μL). The resultant aqueous solution was subjected to reversed-phase LC/MS (column: Waters Symmetry 4.6 × 150 mm, 3.5 µm, flow rate 0.8 mL/min) in three different gradients.

Gradient 1 for iSer (mobile phase CH_3_CN + 0.04% formic acid /H_2_O + 0.04% formic acid, using a linear gradient from 5% to 20% CH_3_CN in 10 min and then from 20% to 35% CH_3_CN in 25 min): the retention time was 23.9 min for l-iSer.

Gradient 2 for Asp (mobile phase CH_3_CN + 0.04% formic acid/H_2_O + 0.04% formic acid, using a linear gradient from 5% to 30% CH_3_CN in 10 min and then from 30% to 50% CH_3_CN in 30 min): the retention time was 17.0 min for d-Asp.

Gradient 3 for Dap: mobile phase CH_3_CN + 0.04% formic acid/H_2_O + 0.04% formic acid, using a linear gradient from 5% to 10% CH_3_CN in 5 min and then from 10% to 35% CH_3_CN in 25 min): the retention times was 15.5 min for l-Dap.

Retention times for the derivatized amino acids standards were as follows: gradient 1 (23.2 min for d-iSer and 23.8 min for the l-iSer); gradient 2 (16.2 min for l-Asp and 17.0 min for d-Asp), and gradient 3 (15.5 min for l-Dap and 16.6 min for the d-Dap).

Absolute Configuration of **2**. First, 0.3 mg of seribunamide A was hydrolyzed in 0.4 mL of 6 N HCl, 110 °C for 15 h. The excess aqueous HCl was removed under a N_2_ stream, and a solution of 400 µg of l-FDAA (N-(3-fluoro-4,6-dinitrophenyl)-l-alanine-amide) in acetone (160 µL), H_2_O (100 µL), and NaHCO_3_ 1N (50 µL) was added. The vial was heated at 40 ºC for 1 h, and the contents were neutralized with 2N HCl (20 µL) after cooling to room temperature. The resulting solution was dried in vacuum and reconstituted in H_2_O (600 μL) before being analyzed by HPLC-MS using two different methods.

Ile was analyzed using Lux Cellulose-4, 5 µm, flow 1 mL/min, H_2_O/AcN + 0.04%TFA isocratic 65:35 in 60 min. The retention time of the l-FDAA amino acid in the hydrolysate of 2 was established as l-Ile 36.1 min. Retention times for the derivatized amino acids standards were as follows: l-allo-Ile 31.7 min, l-Ile 36.1 min, d-*allo*-Ile 38.1 min, and d-Ile 51.4 min.

Pro, NMeLeu, and Glu were analyzed using Symmetry 4.6 × 150 mm, 3.5 µm, flow 0.8 mL/min, H_2_O+ 0.04%TFA/CH_3_CN + 0.04%TFA from 20% to 50% in 30 min. The retention time of the l-FDAA amino acids in the hydrolysate of 2 were established as l-Glu 14.0 min, l-Pro 16.8 min, and NMe-l-Leu 28.5 min. Retention times for the derivatized amino acids standards were as follows: l-Glu 14.1 min, d-Glu 15.2 min, l-Pro 16.8 min, d-Pro 14.01 min, NMe-l-Leu 28.5 min, and NMe-d-Leu 30.1 min.

### 3.5. Biological Activity

The cytotoxic activity of **1** and **2** was tested against four human tumor cell lines, lung (A-549), colon (HT-29), breast (MDA-MB-231), and pancreas PSN-1, and both compounds displayed a GI_50_ > 1.2 × 10^−5^ M in all cell lines. Since these compounds did not show cytotoxicity, we further evaluated an anticancer response through other targets. Thus, compound **1** was further tested for the capacity to inhibit the enzyme topoisomerase I, but it showed no inhibition of the enzyme at 1.0 × 10^−5^ M, and therefore, it was not considered active in inhibiting this enzyme. Likewise, compounds **1** and **2** were unable to impair the interaction between the programmed cell death protein PD-1 and its natural ligand PDL1 as demonstrated by their lack of effect in a cell-based assay whose final readout was dependent on the interaction between the two proteins (Table 3).

## 4. Conclusions

In summary, two new peptides bearing unprecedented halogenated moieties, haloirciniamide A (**1**) and seribunamide A (**2**), were isolated from a marine sponge belonging to the *Irnicia* genus, which was selected for further studies. The sample was collected around the Thousand Islands (Indonesia) by the Pharmamar expedition team in collaboration with the Research Center for Oceanography, Indonesian Institute of Sciences (RCO-LIPI). The planar structures of the novel compounds were determined by a combination of extensive NMR and HPLC-MS experiments. The absolute configuration was achieved by Marfey’s analysis after acid hydrolysis. Cytotoxic activity in the four cancer cell lines tested was not observed for **1** and **2**, as in the case of gunungamide A and cyclocinamide B. In addition, an anticancer immune response was also evaluated, and neither compound was able to impair PD1–PDL1 interaction. Moreover, compound **1** failed to inhibit the enzyme topoisomerase I. Unfortunately, the amount of the compounds isolated was not enough for further assays. This work is the first example of the isolation and structural elucidation of novel compounds with unique structural features from an *Ircinia* sponge, which highlights this gender and its microbiota as a distinctive source of novel structures.

## Figures and Tables

**Figure 1 marinedrugs-18-00396-f001:**
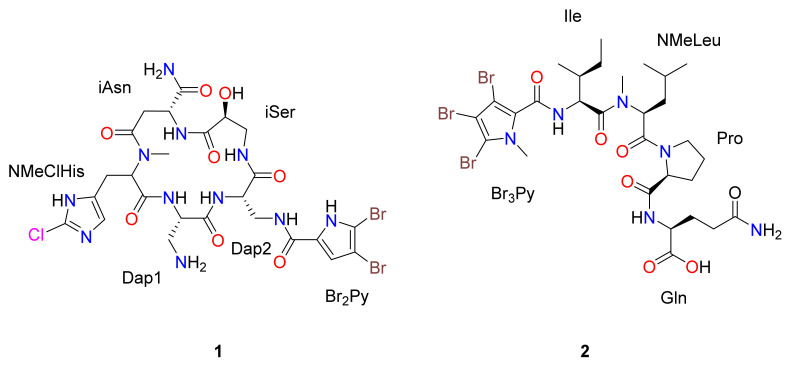
Chemical structures of the compounds **1** and **2** isolated from *Ircinia* sp.

**Figure 2 marinedrugs-18-00396-f002:**
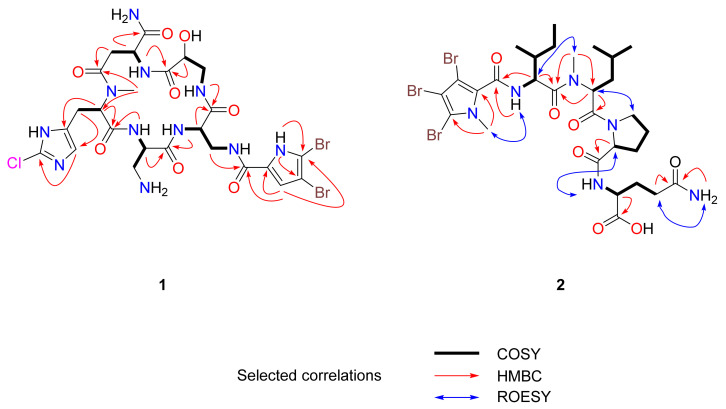
Selected key COSY (bold), HMBC (red), and ROESY (blue) correlations for **1** and **2**.

**Figure 3 marinedrugs-18-00396-f003:**
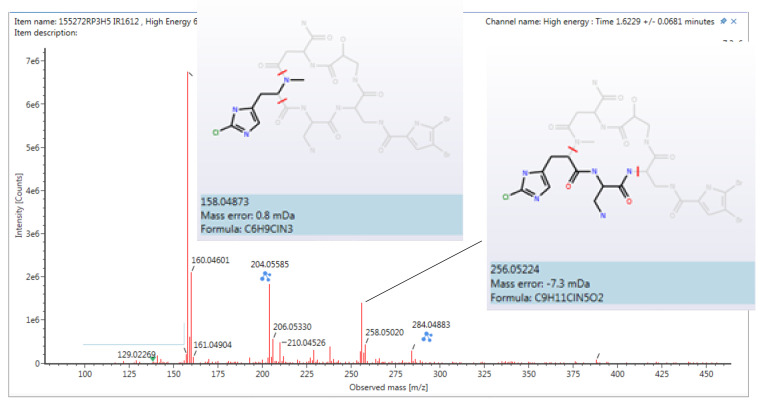
Fragment found for **1** by QTOF.

**Table 1 marinedrugs-18-00396-t001:** NMR spectroscopy data for **1** (^1^H NMR MHz, ^13^C NMR 125 MHz).

	Pos	δ_C_, Mult *^a^*	δ_H_, Mult (*J* in Hz) *^a^*	δ_C_, Mult *^b^*	δ_H_, Mult (*J* in Hz) *^b^*
*Dap1*	1	169.5, C		172.3, C	
	2	50.3, CH	4.23 ddd (6.2, 6.2, 2.8)	52.2, CH	4.54 dd (5.6, 2.9)
	3	49.2, CH_2_	2.87 d (13.9, 6.2)	50.5, CH_2_	3.17 dd (14.6, 5.6)
			3.13 d (13.9, 2.8)		3.54 d (14.2)
	NH		9.11 d (6.2)		8.97 d (6.6)*
	NH_2_				
*NMeClHis*	1	169.0, CO		170.9, CO	
	2	65.6, CH	3.99 dd (10.6, 3.9)	67.0, CH	4.06 dd (10.1, 4.5)
	3	24.9, CH_2_	3.05 m	25.9, CH_2_	3.19 dd (15.2, 10.1)
					3.30 dd (15.2, 4.5)
	4	110.5, C		135.6, C	
	5	109.5, CH	6.76 s	120.8, CH	6.96 s
	6	128.2, C		131.0, C	
	NMe	39.6, CH_3_	2.84 s	40.4, CH_3_	3.02 s
*iAsn*	1	173.5, C		175.8, C	
	2	35.0, CH_2_	2.80 dd (16.6, 2.7)	36.2, CH_2_	3.02 m
			3.09 dd (16.6, 5.8)		3.27 m
	3	48.4, CH	4.58 ddd (8.0, 5.8, 2.7)	50.3, CH	4.84 m
	4	172.2, CO		173.8, CO	
	NH		7.13 d (8.0)		7.61 d (6.3) *
	NH_2_				
*iSer*	1	171.9, C		173.3, C	
	2	67.8, CH	4.13 dd (9.0, 4.2)	69.5, CH	4.40 dd (9.5, 4.0)
	3	42.8, CH_2_	2.75 ddd (9.0, 9.4, 5.4)	44.2, CH_2_	3.00 m
			3.47 m		3.75 dd (13.2, 4.0)
	NH		8.23 t (5.4)		8.25 s *
*Dap2*	1	170.5, C		172.8, C	
	2	51.5, CH	4.49 ddd (9.2, 9.2, 6.3)	53.4, CH	4.84 m
	3	40.1, CH_2_	3.22 m	41.6, CH_2_	3.57 dd (13.8, 8.6)
			4.04 ddd (12.9, 6.3, 6.3)		4.22 dd (13.8, 5.4)
	NH-1		7.72 d (9.2)		7.98 d (9.5) *
	NH-2		7.20 t (6.3, 6.3)		7.37 t (6.0) *
*Br_2_Py*	1	158.6, CO		161.4, CO	
	2	118.0, C		120.2, C	
	3	110.4, CH	6.30 d (2.7)	112.0, CH	6.15 s
	4	96.9, C		99.4, C	
	5	123.2, C		124.2, C	
	NH		12.68 d (2.7)		12.03 s

*^a^* In DMSO-d_6_. *^b^* In CD_3_OD (* CD_3_OH).

**Table 2 marinedrugs-18-00396-t002:** NMR spectroscopy data for **2** (^1^H NMR 500 MHz, ^13^C NMR 125 MHz).

	Pos	δ_H_, Mult (*J* in Hz) ^a^	δ_C_, Mult ^a^	δ_H_, Mult (*J* in Hz) ^b^	δ_C_, Mult ^b^
*Br_3_Py*	1	-	161.8, CO	-	159.1, CO
	2	-	128.6, C	-	128.5, C
	3	-	102.7, C	-	100.7, C
	4	-	101.3, C	-	93.4, C
	5	-	110.8, C	-	107.9, C
	NMe	3.76, s	36.7, CH_3_	3.61, s	35.7, CH_3_
*Ile*	1	-	174.3, CO	-	171.4, CO
	2	4.84, m	55.7, CH	4.65, dd, 8.35, 8.5	53.7, CH
	3	1.96, m	38.0, CH	1.87, m	35.8, CH
	4	1.73, m, 1.22, m	25.9, CH_2_	1.57, m; 1.21, m	24.2, CH_2_
	5	0.95, t, 7.4	11.2, CH_3_	0.83, t, 7.4	10.7, CH_3_
	6	0.99, d, 6.8	15.7, CH_3_	0.85, d, 6.9	15.1, CH_3_
	NH	8.24, d, 8.1	-	8.56, d, 8.2	-
*NMeLeu*	1	-	171.8, CO	-	168.8, CO
	2	5.51, dd, 10.3, 4.7	54.3, CH	5.35, dd, 10.1, 4.3	51.7, CH
	3	1.77, m; 1.61, m	38.0, CH_2_	1.59, m; 1.42, m	36.7, CH_2_
	4	1.56, m	25.7, CH	1.43, m	23.9, CH
	5	0.97, d, 6.2	23.6, CH_3_	0.87, d, 6.2	23.1, CH_3_
	6	0.93, d, 6.1	22.3, CH_3_	0.83, d, 6.2	21.8, CH_3_
	NMe	3.21, s	31.8, CH_3_	3.04, s	30.5, CH_3_
*Pro*	1	-	174.5, CO	-	171.4, CO
	2	4.41, m	61.6, CH	4.31, dd, 8.3, 4.1	59.1, CH
	3	2.23, m; 2.00, m	30.5, CH_2_	2.02, m; 1.80, m	28.9, CH_2_
	4	2.08, m; 1.92, m	26.0, CH_2_	1.90, m; 1.77, m	24.4, CH_2_
	5	3.75, m; 3.69, m	48.8, CH_2_	3.53, m; 3.50; m	46.7, CH_2_
*Gln*	1	-	174.7, CO_2_H	12.53, brs	173.3, CO_2_H
	2	4.41, m	52.8, CH	4.10, ddd, 8.6, 8.5, 5.2	51.5, CH
	3	2.27, m; 1.92, m	30.5, CH_2_	1.93, m; 1.74, m	27.0, CH_2_
	4	2.41, m; 2.32, m	32.6, CH_2_	2.15, m; 2.11, m	31.3, CH_2_
	5	-	177.9, CONH_2_	7.19, s; 6.77, s	173.5, CONH_2_
	NH	-	-	8.08, brs	-

*^a^* In CD_3_OD. *^b^* In DMSO-*d_6_*.

**Table 3 marinedrugs-18-00396-t003:** % PD-1 and TOPO-I inhibition for compounds **1** and **2.** PD-1: programmed cell death protein.

Compound	Target	%Inhibition at 1 × 10^−5^ M
**1**	Top-I	3
**1**	PD-1	0.3
**2**	PD-1	−3.5

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
