# Peer review of "Unique Polyhalogenated Peptides from the Marine Sponge Ircinia sp."

_marinedrugs, 2020, doi:10.3390/md18080396_

Round 1

Reviewer 1 Report

Excellent basic biochemistry work.  Please consider to revise the objective statement, and also add some information bout the importance of these peptides. What are the limitation of your work any possible future work?!  Good luck, I like your novel idea.

Author Response

Thank you for your kind comments. At line 57, I have included the following new text: In the course of our screening program to isolate novel compounds with antitumor properties from marine sources, the organic extract of an Ircinia specimen collected off the coast of Thousand Islands showed hints of activity, and although the fractions did not confirm cytotoxicity, the chromatographic profiles along with the mass spectra showed peaks interesting enough for us to purify them.

The importance of these peptides remains to be seen. The biological test results that we have reported are only an initial evaluation and the biological importance of these peptides will only be known once a much larger range of tests have been conducted, that we do not have access to, in order to evaluate the possible activity in many more therapeutic areas.

With regard to the limitation of our work, the amount of compound isolated was just enough for the biological tests and structural elucidation. We did not have enough compounds left to continue evaluating them.

At line 277, I have included the following new text: Unfortunately, the amount of the compounds isolated was not enough for further assays.

Reviewer 2 Report

This study described two new bromopyrrole peptides and have been isolated from an Indonesian marine sponge of the genus Ircinia collected in Thousand Islands of Indonesia. Authors showed the very unique structure of both compounds with 2D NMR spectroscopy and mass spectrometry although both compounds did not inhibit cell growth, the enzyme topoisomerase I or impair PD1-PDL1 interaction in human tumor cell lines.  The article is well written and is enough quality to be published into Marine drug.  However, authors did not mention about a permission to take a sponge in Methods. I am afraid this point.

Author Response

Thank you for your valuable comments. I attach here the letter of research permit. Please, keep in mind that this is confidential information and should not be shared or made public.

Reviewer 3 Report

Fernández et al. present biological feature of two compounds derived from Marine Sponge Ircinia. While componds extraction is well explained, not the same is to compound's biological feature.

For example, in introduction the authors do not specify what are searching with PD-1 and TOPOI screening. Moreover, results are weak to talk about noone cytotoxic activity in human cancer cell lines analyzed. 

Finally, in my opionin I don't think that a negative result is not a result but I also think that a novelty about the two compounds should be postulate. These new ideas are not neither in abstract or discussion.

Author Response

Thank you for your valuable comments. We decided to study this sample because its organic extract showed hints of cytotoxic activity. However, after fractionation, the activity was not confirmed. Despite this, the chromatographic profile of this sample was very interesting and we decided to purity and isolate several of the peaks. In this way, we also increase our pure compound data bases. As part of this work, we checked again the cytotoxicity of the pure compounds. Since they were inactive, we tested the compounds in other screens that we have access to at PharmaMar.

As I mentioned to reviewer 1, I have included at line 57 some new text about the decision to study this sample.

At line 74, I have also included that the PD1 and TOPO I screening are other test related to anticancer activity.

At line 257, I have included that both compounds displayed a GI50 >1.2 E-5 M in all cell lines. Since these compounds did not show cytotoxicity, we further evaluated an anticancer response through other targets.

At line 274, I have mentioned that these peptides are not active as in the case of other reported halogenated peptides. In addition, an anticancer immune response was also evaluated and neither compound was able to impair PD1-PDL1 interaction. Moreover, compound 1 failed to inhibit the enzyme topoisomerase I.

At lines 14, 61, 73, 267 and 278 is mentioned the novelty of these structures.

At line 279, I declared that Ircinia and its microbiota are a new and distinctive source of novel structures, because I think that the microorganisms are the real source of these compounds, although this cannot be demonstrated from the current work because we just isolated them from an extract of the whole organisms.

Reviewer 4 Report

I would like to thank the authors for what appears to be a quality paper.  They have identified unique peptides that warrant further study and have provided some limited data on biological activity.  I have only a few small critiques from style.

Firstly, Figure 3 is a bit blurry in terms of the structures represented on it.  Is it possible to please provide a higher resolution image? 

Secondly, could the authors please qualify their statements towards the end about activity.  This reviewer agrees and thinks’ the authors assessment about the lack of activity is correct.  It would be good to offer a reference on a known, standard peptide tested against the cell lines mentioned.  This might give the readers a little bit more of a point of reference and something to think about for possibly improving these peptides down the line.

Overall, not many changes seem to be made, the paper flows logically and the data support the conclusions.

Author Response

Thank you for your valuable comments. We have provided a higher resolution image (line 128).

Due to the high novelty of these peptides, it is precarious to offer a comparison with other reported peptides. However, Gunungamide A with a halogenated pirrole ring in its structure also showed no activity at 15 μM in any of the four human tumor cell lines tested (A-549, HT-29, MDA-MB-231 and PSN-1)

On the other hand, Cyclocinamide A (3) was reported to be highly cytotoxic while cyclocinamide B (2) had no cytotoxicity against HCT-116. Interestingly, in spite of their close structural resemblance, the presence or not of a chlorine atom is crucial to their activity.

In the manuscript, I have included the text highlighted in yellow (line 274):

Cytotoxic activity in the four cancer cell lines tested was not observed for 1 and 2, as in the case of gunungamide A and cyclocinamide B.

Round 2

Reviewer 3 Report

Dear authors,

thank you for your comments and for the changes made in improving paper's biological content. 

Reading comment of collegues, certanly more competent than me in biochemistry, I have understood that compound's biochemical structure is the strong part of the paper. Thanks to the insights you made in the functional and biological part I accept the paper in the present form.